# Evaluation of Spore Acquisition, Spore Production, and Host Survival Time for Tea Shot-Hole Borer, *Euwallacea perbrevis,* Adults after Exposure to Four Commercial Products Containing *Beauveria bassiana*

**DOI:** 10.3390/insects14090726

**Published:** 2023-08-24

**Authors:** Alejandra V. Chavez, Emily B. Duren, Pasco B. Avery, Marco Pitino, Rita E. Duncan, Luisa F. Cruz, Daniel Carrillo, Liliana M. Cano, Ronald D. Cave

**Affiliations:** 1Entomology and Nematology Department, Indian River Research and Education Center, University of Florida, 2199 South Rock Road, Fort Pierce, FL 34945, USA or alejandravelez26@hotmail.com (A.V.C.); dureneb@mail.irsc.edu (E.B.D.); rdcave@ufl.edu (R.D.C.); 2Plant Pathology Department, Indian River Research and Education Center, University of Florida, 2199 South Rock Road, Fort Pierce, FL 34945, USA; marco.pitino@agrosource.net (M.P.); lmcano@ufl.edu (L.M.C.); 3Entomology and Nematology Department, Tropical Research and Education Center, University of Florida, 18905 S.W. 280 Street, Homestead, FL 33031, USA; ritaduncan.florida@gmail.com (R.E.D.); luisafc1@gmail.com (L.F.C.); dancar@ufl.edu (D.C.)

**Keywords:** ambrosia beetle, entomopathogenic fungi, spore acquisition, spore production, commercial products, avocado bark plug bioassay, *Euwallacea* nr. *fornicatus*

## Abstract

**Simple Summary:**

The tea shot-hole borer (TSHB) is an invasive ambrosia beetle that vectors several fungal plant pathogens that cause *Fusarium* branch dieback in avocado trees. This study assessed the potential of four commercial products containing the entomopathogenic fungus *Beauveria bassiana* (*Bb*) for managing and mitigating the spread of the TSHB. The formulated fungal products tested were BioCeres WP, BotaniGard WP, BotaniGard ES and Velifer ES. Acquisition of spores by adult beetles dipped in the product suspensions with 2.5 ± 0.1 × 10^6^ spores/mL was assessed. Survival time of beetles after residual exposure to the products in an in vivo avocado bark plug bioassay was also determined. Controls consisted of water only, BotaniGard ES and Velifer ES suspension supernatants with the spores removed. Production of *Bb* spores on beetles was assessed. Beetles exposed to Velifer ES and BotaniGard ES acquired significantly more spores than those exposed to the other fungal products. Beetles exposed to Velifer ES and BotaniGard ES died faster (6–8 days) compared to beetles treated with the other fungal products (10–11 days) and controls (12 days). Percentage mycosis was highest with beetles exposed to Velifer ES (63%). Spore production on cadavers of beetles dipped in Velifer ES was the highest among all treatments, whereas it was lowest on cadavers of beetles dipped in BotaniGard ES. All fungal products in this laboratory study demonstrated potential for use in management of the TSHB, especially Velifer ES. These *Bb*-based fungal products should be tested under field conditions to confirm these laboratory results.

**Abstract:**

*Euwallacea perbrevis*, the tea shot-hole borer (TSHB), is an invasive ambrosia beetle that vectors several fungal pathogens that cause *Fusarium* branch dieback in avocado trees in southern Florida. This study assessed the potential of four commercial products containing the entomopathogenic fungus *Beauveria bassiana* (*Bb*) for managing adult TSHB beetles. Formulated products containing *Bb* strains to which adult beetles were exposed were BioCeres WP, BotaniGard WP, BotaniGard ES, and Velifer ES. Controls consisted of water only and BotaniGard ES and Velifer ES supernatant with spores removed. Acquisition of spores by adult beetles dipped in product suspensions with 2.5 ± 0.1 × 10^6^ spores/mL was assessed. Survival time of beetles after residual exposure to the *Bb*-based products in an in vivo avocado bark plug bioassay was determined. Production of *Bb* spores on beetles after being dipped in product suspensions and placed in a moistened bark-plug assay with water only was assessed. Significantly more spores were acquired by beetles exposed to Velifer ES and BotaniGard ES than beetles exposed to the other fungal products. Beetles exposed to Velifer ES and BotaniGard ES died faster (6–8 days) compared to beetles dipped in the other fungal products (10–11 days) and controls (12 days). Percentage of mycosis was highest with beetles exposed to Velifer ES (63%). Spore production on cadavers of beetles dipped in Velifer ES (20 ± 6.4 × 10^5^ spores/cadaver) was the highest among all treatments, whereas it was the lowest on cadavers of beetles dipped in BotaniGard ES (1 ± 0.2 × 10^5^ spores/cadaver). All *Bb*-based products, especially Velifer ES, demonstrated potential to manage TSHB populations under laboratory conditions. These *Bb*-based fungal products should be tested under field conditions to confirm these laboratory results.

## 1. Introduction

Ambrosia beetles live inside excavated galleries of host trees and culture symbiotic fungi that serve as their primary food source [1,2]. The cultured symbiotic fungus inside the galleries also produces phytotoxic secondary metabolites that destroy the functional xylem, thereby interrupting the transport of water and nutrients and causing the health of the tree host to be negatively impacted [3]). The tea shot-hole borer (TSHB), *Euwallacea perbrevis* Schedl (previously *E*. nr. *fornicatus* prior to the taxonomic revision by Smith et al. [4] (Coleoptera: Curculionidae: Scolytinae)) is an ambrosia beetle that cultures and vectors several fungal plant pathogens that cause *Fusarium* stem dieback disease in host trees [5,6,7,8]. These plant hosts include mango (*Mangifera indica* L., (Anacardiaceae)), swampbay (*Persea palustris* (Raf.) Sarg., (Lauaraceae)), and royal poinciana (*Delonix regia* (Boj. ex. Hook) Raf, (Fabaceae)) [9,10,11]. However, in 2010, TSHB was observed attacking healthy avocado trees, *Persea americana* Mill (Lauraceae), in southern Florida [11,12]. Since this discovery, TSHB has spread throughout the commercial avocado production region centered around the Homestead area in Florida [10,13]. The physical sign of damage by the TSHB in avocado trees is the presence of white sugary exudates, “volcanoes”, around the areas of beetle attack, which results in dead twigs and branches [6].

Until recently, management strategies for the ambrosia beetles attacking avocado trees relied solely on chemical control and sanitation practices [10,14]. However, insecticides are only effective when the beetles are outside the tree, yet most of the pest’s life cycle occurs inside the tree. Therefore, conventional insecticides are only recommended to suppress active ambrosia beetle populations after removing and chipping infested trees. Beneficial entomopathogenic fungi (EPF) have been demonstrated to be efficacious for infecting and managing the ambrosia beetles *Xyleborus bispinatus* Eichhoff (Coleoptera: Curculionidae: Scolytinae) and *Xyleborus volvulus* (Fabricus, 1775) attacking avocado trees [15,16]. The susceptibility of the TSHB to infection by EPF has not been assessed and is the focus of our study.

The commercially available fungal biopesticide BotaniGard ES^®^ containing *Beauveria bassiana* (*Bb*) spores is already adopted and employed by most avocado growers as part of their integrated pest management strategy for mitigating ambrosia beetles. In earlier studies, only BotaniGard ES^®^ (*Bb* strain GHA) was assessed against ambrosia beetles [15,16,17]. However, more formulated *Bb*-based products with different strains have been registered and are now commercially available. Therefore, the standard formulated *Bb*-based product (BotaniGard ES^®^) and the new commercially available *Bb*-based formulated products need to be tested for their pathogenicity against adult TSHB to determine if any of the new fungal products are more pathogenic and cost effective than the standard.

Biotic (e.g., host density, developmental stage) and abiotic (e.g., humidity, temperature, photoperiod) factors can regulate epizootics in insects [18,19,20,21]. One biotic factor is the epizootic potential, which depends not only on the ability of the fungi to sporulate on cadavers, but also their ability to horizontally transmit the infective propagules to healthy insects, a process known as secondary cycling [22,23,24,25,26,27,28,29]. TSHB adults could acquire *Bb* spores while roaming on the bark before boring into the tree to lay eggs in constructed galleries. Any progeny that is produced should potentially be killed via horizontal transmission of the spores produced on the cadaver. Production of these spores is important for achieving long-term control and can potentially increase the horizontal transmission efficiency of the fungal entomopathogen. In our study, the spore production on the adult TSHB cadavers was assessed as a biotic parameter for determining the epizootic potential of each *Bb* strain contained in the formulated product.

The objectives of our study were to determine (1) the number of conidia acquired per adult TSHB for each *Bb*-based formulated product; (2) the survival time of adult TSHB after exposure to *Bb*-based formulated products; and (3) the epizootic potential of an adult TSHB cadaver by quantifying the spore production of each *Bb* strain.

## 2. Materials and Methods

### 2.1. Beetles

Adult TSHB beetles were obtained from a laboratory colony maintained by following the procedures described by Menocal et al. [30,31]. All adult females in the bioassays were used within 2 days of collection. To reduce bias in the selection of beetles for the bioassays among the different treatments and controls, dishes containing beetles were stacked randomly, and then 2–3 actively moving beetles were randomly chosen within the dish. In addition, the dishes were shuffled in the stack after removing the selected beetles from each dish.

### 2.2. Fungal Products and Controls

The formulated mycoinsecticides containing the same or different *Bb* strains exposed to the TSHB adults were BioCeres^®^ WP (active ingredient (ai): *Bb* strain ANT-03; BioSafe Systems, East Hartford, CT, USA); BotaniGard^®^ 22WP (ai: *Bb* strain GHA; Laverlam International, Butte, MT, USA); BotaniGard^®^ ES (ai: *Bb* strain GHA; Laverlam International, Butte, MT, USA); and Velifer^®^ ES (ai: *Bb* strain PPRI 5339; BASF Corporation, Research Triangle, Park, NC, USA) (Table 1). Each product, whether a wettable powder (WP) or an oil-emulsified suspension (ES), was prepared according to the label instructions. The resulting stock suspensions were diluted to ~2.4 × 10^6^ (range: 2–3 × 10^6^) spores/mL, and the concentration was confirmed using a Neubauer improved Incyto C-chip plastic disposable hemocytometer (INCYTO, Chungnam-do, Republic of Korea) viewed under a light microscope (Leica Microsystems Inc., Buffalo Grove, IL, USA) at 400X. Beetles in the control groups were exposed to water only for comparison to the beetles treated with WP-formulated products. For comparison with the ES-formulated products, TSHB adults were exposed to a *Bb* suspension after filtration of the spores, leaving only the formulated filtrate. The filtrates were prepared as follows: (1) 10 mL of the original suspension spores was poured into a 30 mL EXCELint^®^ disposable plastic syringe (Excelint International, Co., Los Angeles, CA, USA) without the plunger; (2) the plunger was inserted into the syringe tube, and the suspension was filtered through a 0.45 µm Corning^®^ sterile syringe filter (Corning, Inc., Corning, NY, USA). The clean filtrate was analyzed using a C-chip hemocytometer to confirm the absence of spores. Spore viability via percentage germination was determined with the technique described by Avery et al. [15] for all fungal products used in the experiments. The spore viability for all *Bb*-based products was >95%.

### 2.3. Spore Acquisition by Adult TSHB

The protocol for determining the acquisition of spores was described by Avery et al. [15]. The only modification to the protocol was that the beetles were placed individually in a safe-lock microcentrifuge tube (1.5 mL) with a snap cap containing 200 µL of Tween 80 (0.01%) instead of Triton X-100. The snap cap was closed, and the tubes were vortexed for 15 s to remove the spores from the beetle and allowed 1 min for the spores to settle to the bottom of the tube. One aliquot (10 µL) of each suspension removed from the bottom of the tube was placed in the C-Chip hemocytometer in 2 different sides (A and B), and the number of spores/mL per side was counted and averaged per beetle. Five beetles were used per treatment, and the experiment was conducted three times on separate occasions for a total of fifteen beetles per treatment.

### 2.4. Adult TSHB Survival Time

The survival time of adult TSHB in days after exposure to the product formulations in the in vivo avocado bark plug bioassays was determined following the protocol of Zhou et al. [16]. The mean ± SE spore concentration for each formulation applied to the bark plug was 2.4 ± 0.1 × 10^6^ spores/mL. Beetles were examined 21 days after exposure for mycosis, and each fungal phenotype was verified (Table 1). The percentage of beetles that mycosed was recorded per treatment in each trial. Based on availability, a total of 20, 20 and 10 beetles were used per treatment for trials 1, 2 and 3, respectively, for a total of fifty beetles per treatment on three separate occasions.

### 2.5. Spore Production on TSHB Cadavers

Contamination of TSHB adult beetles followed the protocol described for dipping in Avery et al. [15]. However, to measure spore production after contamination and death, the beetles were individually placed inside an in vivo avocado bark plug assay tube, except the bark plug inside the tube was only moistened with 75 µL of sterile distilled water. Each beetle was monitored daily for mortality as per the protocol described by Zhou et al. [16]. Dead beetles with no signs of mycosis were removed and surface sterilized according to the technique described by Lacey and Brooks [32]. Dead and mycosed beetles were carefully removed from the bioassay with a moistened camel-hair brush and not surface sterilized. Each beetle, whether surface sterilized or mycosed, was placed in separate Petri dishes (50 mm Ø) with a moistened Whatman™ filter paper circle (42.5 mm Ø), and the dishes were sealed with Parafilm. Each dish was labeled according to the date of beetle death and placed inside a growth chamber set at 25 °C with a 14 h photophase for an additional 21 days to allow for maximum sporulation. The 21 days for mycosis and spore production for each beetle cadaver under optimum conditions allowed for those strains that were less virulent to sporulate.

By using a sterile moistened camel-hair brush, mycosed cadavers at 21 days were carefully placed individually in safe-lock microcentrifuge tubes (1.5 mL) containing 200 µL of Tween 80 (0.01%). The procedure described above was followed for spore acquisition. The percentage of beetles that mycosed was recorded. Ten beetles (5 beetles per group) were assessed per treatment per trial, and the experimental trial was conducted twice on separate occasions for a total of twenty beetles assessed per treatment.

### 2.6. Statistical Analyses

The mean number of spores acquired and produced as well as days to death were statistically analyzed for significant differences by using one-way ANOVA (*p* < 0.05). The percentages of mycosed beetles per treatment from both trials were combined, averaged, and statistically analyzed after the data were square-root-arcsine transformed. If ANOVA was significant using the transformed data, post hoc mean values per beetle treatment were separated and compared using the LSD test (*α* = 0.05). Statistical comparison for significance for the number of spores acquired after dipping and the production of spores on the cadavers among *Bb*-formulated products was determined using a Student’s *t*-test (*α* = 0.05). Statistical analyses were conducted using SAS Proc GLM procedures and executed on a PRO_WIN 6.1 platform (SAS 2002–2012; SAS Institute Inc., Cary, NC, USA). Median survival times for 50% of the population (ST_50_) per treatment were compared using a Kaplan–Meier survival analysis followed by a log-rank test (*p* < 0.05) (SAS JMP Pro 16 for Windows 2013).

## 3. Results

### 3.1. Spore Acquisition by Adult TSHB

The mean number of spores acquired by the dipped beetles varied depending on product formulation and *Bb* strain (Figure 1). The mean number of spores on beetles dipped in ES-formulated products suspended in oil was significantly higher (*F*_3,42_ = 17.63; *p* < 0.0001) compared to spore acquisition by beetles dipped in WP-formulated products mixed with water.

### 3.2. Adult TSHB Survival Time

Kaplan–Meier analysis (censored at 21 days) revealed in a log-rank test that there were significant differences (*χ*^2^ = 58.05; *df* = 6; *p* < 0.0001) in the ST_50_ values among treatments. The mean survival time in days for beetles exposed to Velifer ES and BotaniGard ES were significantly shorter (*F*_6,291_ = 7.34; *p* < 0.0001) compared to that of beetles exposed to the control treatments; mean survival times after exposure to BioCeres WP and BotaniGard WP were similar to those of the water control (Figure 2).

The percentages of mycosed TSHB adults exposed to *Bb* in the bark plug bioassays were not significantly different (*F*_3,27_ = 2.13; *p* = 0.05). The percentages ± SE of mycosis for Velifer ES, BotaniGard WP, BotaniGard ES, and BioCeres WP were 58% ± 10.1, 50% ± 6.1, 40% ± 6.0, and 30% ± 4.5, respectively. None of the dead TSHB from the water or supernatant controls mycosed.

### 3.3. Spore Production on TSHB Cadavers

The mean number of spores produced on mycosed TSHB adult cadavers was significantly higher for Velifer ES (*F*_3,47_ = 7.63; *p* = 0.0003) than for the other treatments (Figure 3). Spore production was lowest for beetles treated with BotaniGard ES.

The percentages of mycosed TSHB adults that were dipped in the *Bb*-based products were not significantly different (*F*_3,9_ = 1.14; *p* = 0.3852). The percentages ± SE of mycosis for beetles exposed to BotaniGard ES, Velifer ES, BotaniGard WP, and BioCeres WP were 100% ± 0.0, 90% ± 5.8, 75% ± 15.0, and 70% ± 17.3, respectively.

### 3.4. Comparison of Inoculum Spore Acquisition by Dipping versus Cadaver Sporulation

The mean number of spores produced per adult TSHB cadaver was significantly greater than that of spore acquisition via dipping live beetles for BotaniGard WP (*t*_31_ = −2.08; *p* = 0.0456) and Velifer ES (*t*_30_ = −2.68; *p* = 0.0118) (Figure 4). The mean number of spores produced on the cadavers was significantly lesser for BotaniGard ES (*t*_33_ = 3.19; *p* = 0.0031) compared to that of acquisition by dipping live beetles. No significant difference between the mean numbers of spores acquired (mean ± SD: 0.7 ± 0.78; range: 0.06–1.9) by live beetles or produced (mean ± SD: 9.5 ± 21.3; range: 0.01–84.0) on cadavers was observed for BioCeres WP (*t*_28_ = −1.60; *p* = 0.1207).

## 4. Discussion

This is the first study to demonstrate the efficacy of *Bb*-based products screened against adult TSHB and assess the mean number of *Bb* spores that beetles can acquire under laboratory conditions. In addition, this study lends support to the original finding that *Xyleborus glabratus* Eichhoff, *X. volvulus*, *X. bispinatus*, *Xylosandrus crassiusculus* (Motschulsky), and now *E. perbrevis* are all susceptible to infection by the same GHA strain contained in the *Bb*-based biopesticide, BotaniGard ES. This formulated product is commercially available nationwide and used by avocado growers against ambrosia beetles. The mean production of spores per adult TSHB cadaver was also measured for each product. Indirectly, the quantification of the spores produced on the adult TSHB cadaver is an assessment of the epizootic potential of the *Bb* strain for each formulated product.

### 4.1. Spore Acquisition by Adult TSHB

Pathogenesis and mycosis of the host arthropod by an entomopathogenic fungus begins with the acquisition of spores onto the epicuticle of the susceptible host [33]. In our study, adult TSHB beetles differed significantly in their ability to acquire the spores suspended in the *Bb*-based biopesticides. Avery et al. [15] and Carrillo et al. [17] found that the mean number of spores acquired by other ambrosia beetle adults dipped in different fungal products varied significantly. The authors suggested that the variance in the number of spores acquired per beetle species may be due to physicochemical surface properties of the adult epicuticle. However, in contrast to their studies, only the TSHB was assessed in our study; therefore, the physiochemical surface properties were similar among the beetles used in the experiment. Perhaps the discrepancy in the mean number of spores acquired by the beetles per treatment may be due to the technique we employed, which might not have effectively dislodged the spores from the epicuticle. In our study, adult TSHB were contaminated in individual wells containing the fungal suspension, then vortexed or washed in 0.01% Tween 80 solution, as described by Goettel and Inglis [34]. All beetles were contaminated with the same concentration of spores (2.4 × 10^6^ spores/mL). Goettel and Inglis [34] indicated that spores can be dislodged by vigorous washing in water amended with a surfactant of 0.01% Tween 80 and then using a mechanical shaker (we used a vortexer), although propagules of entomopathogenic fungi may be firmly attached to the insect’s epicuticle. In our study, beetles were vortexed in water amended with 0.01% Tween 80 immediately following their contamination, so our technique of vortexing after contamination should have been sufficient to wash and dislodge the spores from the epicuticle. Goettel and Inglis [34] also noted that the quantity of recovered spores usually represents a conservative estimate of the true spore abundance on the host due to clumping of propagules and/or inadequate detachment from the integument. Therefore, the mean number of spores recovered from adult TSHB beetles per *Bb*-based product is considered a conservative estimate of the true spore acquisition. Although conservative estimates, our mean numbers of spores recovered from beetles can justifiably be compared among the four *Bb*-based formulated products.

The insect epicuticle is typically hydrophobic [35], and the adhesion of spores is determined primarily by hydrophobic interactions on the surface [36]. All the *Bb*-based products tested in our study contain hydrophobic aerial conidia [37,38]; only the formulation and fungal strain differed. It seems that the spores in the wettable powder (WP) formulated products did not adhere as strongly to the beetle epicuticle compared to the emulsifiable suspensions (ES) in oil. For example, BotaniGard WP and BotaniGard ES both contain the GHA strain; yet the beetles acquired significantly more spores when contaminated in the emulsifiable oil product compared to those exposed to the wettable powder formulation. Entomopathogenic fungi in emulsifiable oil have been demonstrated to have enhanced the adhesion of spores to the arthropod cuticle through hydrophobic interactions between the spore and the epicuticle [39]. Prior et al. [40] demonstrated that *Bb* spores in oil formulations adhere significantly better to the insect epicuticle of the cocoa weevil, *Pantorhytes plutus* (Oberthür), compared to the water formulation. Therefore, in our study, the variation in the mean numbers of spores acquired per beetle per *Bb*-based product should be due to differences in the ability of the product formulation type to attach the spores to the beetle epicuticle.

### 4.2. Adult TSHB Survival Time

The mean survival time of the adult TSHB beetles per formulation varied similarly to their spore acquisition. This variation in survival time is directly related to the successful adherence of the *Bb* spores per strain per formulation type to the beetle epicuticle and then the subsequent infection process, followed by pathogenicity of the susceptible beetle host. Individual beetles exposed to either Velifer ES or BotaniGard ES residual propagules in the in vivo avocado bark plug bioassays died 1–2 days faster than those exposed to BioCeres WP or BotaniGard WP. In general, it seems that the spores contained in the WP products did not adhere as firmly to the epicuticle of the beetles compared to the ES products in oil. When ES products are applied as an oil-in-water emulsion, the spores are generally surrounded by oil droplets, which may enhance their adherence to the insect epicuticle and level of efficacy [15]. Batta [41] demonstrated that a significantly higher level of efficacy was obtained using *Bb* spores in an oil emulsion formulation compared to no emulsion when applied against the almond bark beetle, *Scolytus amygdali* Guérin-Méneville. In another study, Prior et al. [40] observed that *Bb* spores in an oil-based formulation adhered significantly better to the epicuticle of the cocoa weevil and enhanced mortality with a lower lethal time value compared to the water formulation under laboratory and field conditions. Yasuda et al. [42] observed that infectivity of the sweet potato weevil, *Cylas formicarius* (Fabricius), to *Bb* spores was enhanced when 10% corn oil was added to the suspension. Overall, oil-based formulations of EPF-based biopesticides have been reported to increase the adherence of propagules to the insect integument by (1) enhancing the laminar characteristics of the inoculum and subsequent penetration of the insect epicuticle, (2) protecting the propagules from being degraded from ultraviolet light radiation, and (3) augmenting infection under low humidity conditions [43].

Other physiochemical aspects that may inhibit or enhance germination and penetration of the spores include cuticle density or compounds on the insect integument [44,45]. Also, the lack of nutrients in the sclerotized beetle cuticle is a limiting factor in fungal growth and development [46]. Other physical or chemical variables defining the interactions at the epicuticular barrier between the EPF and insect that ultimately lead to either successful mycosis by the insect pathogen or successful defense by the host have been thoroughly reviewed by Oritz-Urquiza and Keyhani [47]. Our data suggest that the formulation of each EPF product can affect the adherence of spores to the beetle epicuticle in an in vivo avocado bark plug bioassay, which could have a subsequent positive effect on successful mycosis and production of spores, which can promote secondary cycling as well as potential horizontal transmission via the female sporulating cadaver [17,24,25].

### 4.3. Spore Production on TSHB Cadavers

In our study, each *Bb*-contaminated beetle was allowed ample time (21 days) under optimal environmental conditions (25 °C and ~100% RH) for each fungal strain to effectively adhere, penetrate the epicuticle, and demonstrate its infectivity, virulence, and overall pathogenicity. Infected beetles that mycosed followed by production of the *Bb* spores on the beetle cadaver were used as an indicator of the epizootic potential; significant variation was detected among the formulated products. The mean number of spores produced by the mycosed adult TSHB beetle cadaver was highest for Velifer ES. This variability in spore production for each formulated product is likely related to the genetic variation of the *Bb* strains as well as the number of viable spores that successfully adhered to and penetrated the epicuticle.

### 4.4. Comparison of Beetle Conidia Acquisition versus Cadaver Sporulation

There was a significant increase in the mean number of spores produced per adult TSHB beetle cadaver for BotaniGard WP and Velifer ES. For BotaniGard ES, there was a significant decrease in the mean number of spores produced on cadavers compared to that of the acquisition by the living beetle during dipping in the product suspension. Also, for the other *Bb*-based products tested, similar mean numbers of conidia were acquired and produced on the beetle cadaver. Once the spores penetrate the epicuticle via lipase, esterase, chitinase, and protease enzymes and breach the hemocoel of the beetle, toxins such as beauverolides, bassianolides, beauvericins, and isarolides [33] are produced by different strains of *Bb*. Production of the different enzymes and genetic characteristics help determine the pathogenicity and virulence of the fungal strain as well as the production of spores on the beetle cadaver. Therefore, in our study, perhaps the *Bb* strain in Velifer ES is more virulent and produces more spores than were acquired. Conversely, BotaniGard ES is less virulent and produced fewer spores than what was acquired. However, Velifer ES and BotaniGard ES acquired similar numbers of spores and the survival times for the TSHB beetles were also similar; only the production of spores differed. The hypothesis about the relationship between virulence and production of spores was not the focus of our study and needs further research to verify the results of this investigation.

The focus of our study was to determine the most virulent *Bb* strain and formulation that produces the most spores per beetle cadaver. Based on the results of this laboratory study, the *Bb* strain in Velifer ES appears to have properties that are similar to or better than the standard fungal biopesticide presently being used by the avocado growers, BotaniGard ES. However, these *Bb*-based biopesticides have only been tested under laboratory conditions and need to be tested under field conditions to verify the results, especially after application on the bark of the trunk of the avocado trees. Before boring into the trunk of the avocado tree, the TSHB female, similar to other ambrosia beetles [48], lands on the tree bark and acquires the spores, previously sprayed on the surface, while it roams to find a place to bore [17]. If the infected adult TSHB female bores into the avocado tree and dies inside the gallery, the spores produced on the cadaver spread to the progeny via horizontal transmission [24,25]. If this horizontal transmission of spores occurs within the galleries, it can cause an epizootic and kill all the eggs and progeny, which ultimately mitigates the spread of phytopathogens vectored by the TSHB. Also, in support of this scenario occurring under field conditions, Selvasundaram and Muraleedharan [49] recently recorded, in a survey conducted throughout India, that several female TSHB beetles infesting tea plants were observed to be mummified by a strain of *Bb*. Considering that these bioassays were conducted under optimum laboratory conditions, the apparent positive effect of the oil-based formulations remains to be confirmed under field conditions.

## 5. Conclusions

All *Bb*-based products evaluated in our in vivo avocado bark plug bioassays demonstrated potential for being used for management of the TSHB beetles. In field experiments, it was revealed that EPF do not prevent female ambrosia beetles from boring into the trees. However, the EPF-infected beetles die inside the trees and mycose without reproducing. Although this boring characteristic was not assessed in our study, beetles after exposure to the *Bb*-based products mycosed and sporulated. *Bb*-based products may persist on the bark of avocado trees and potentially cause epizootics under the proper environmental conditions. Our study demonstrated that TSHB adult beetles are susceptible to infection by the *Bb* strains contained in the fungal products BioCeres WP, BotaniGard WP, BotaniGard ES and Velifer ES and should be able to be managed effectively. According to our study, Velifer ES seems to perform as well as the standard fungal product, BotaniGard ES, presently being applied to avocado trees by growers in Florida. However, Velifer ES needs to be tested under field conditions to confirm the results of our laboratory in vivo study. To improve field efficacy, *Bb*-based products must be integrated into the management strategy for mitigating *Fusarium* branch dieback, laurel wilt disease, and other avocado pests, which includes the use of several fungicides, insecticides, and adjuvants. The main limitation of EPF is they cannot control ambrosia beetles breeding inside the trees. Therefore, the search for better delivery systems that increase the efficacy of EPF, as well as the integration of biofungicides for reducing the spread of *Fusarium* branch dieback and laurel wilt disease, is still ongoing.

## Figures and Tables

**Figure 1 insects-14-00726-f001:**
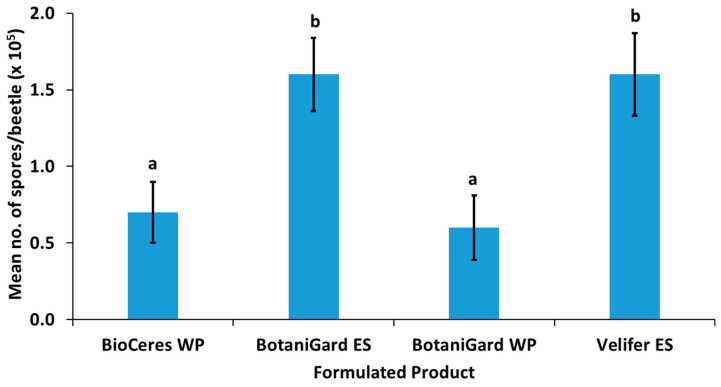
Mean ± SE number of spores acquired per adult TSHB dipped in four formulated entomopathogenic fungal products containing *Beauveria bassiana*. Columns with different letters above the SE bars are significantly different among the treatments (LSD test, *p* < 0.05). WP = wettable powder; ES = emulsifiable suspension. No *B. bassiana* spores were observed on beetles dipped in water + Tween 80.

**Figure 2 insects-14-00726-f002:**
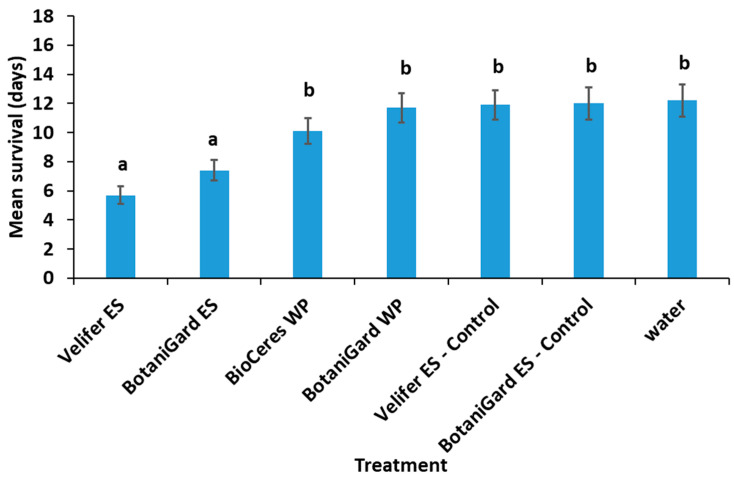
Mean ± SE survival time in days for TSHB adults after exposure to four formulated entomopathogenic fungal products containing *Beauveria bassiana* and three control treatments in an in vivo avocado bark plug bioassay. Columns with different letters above the SE bars are significantly different among the treatments (LSD test, *p* < 0.05). WP = wettable powder; ES = emulsifiable suspension.

**Figure 3 insects-14-00726-f003:**
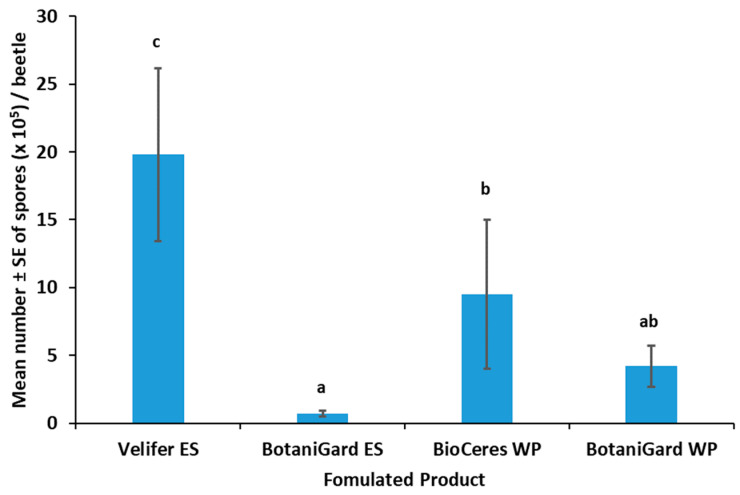
Mean number ± SE of spores per TSHB cadaver mycosed after 21 days with *Beauveria bassiana* from four formulated products. Columns with different letters above the SEM bars are significantly different among the treatments (LSD test, *p* < 0.05). WP = wettable powder; ES = emulsifiable suspension.

**Figure 4 insects-14-00726-f004:**
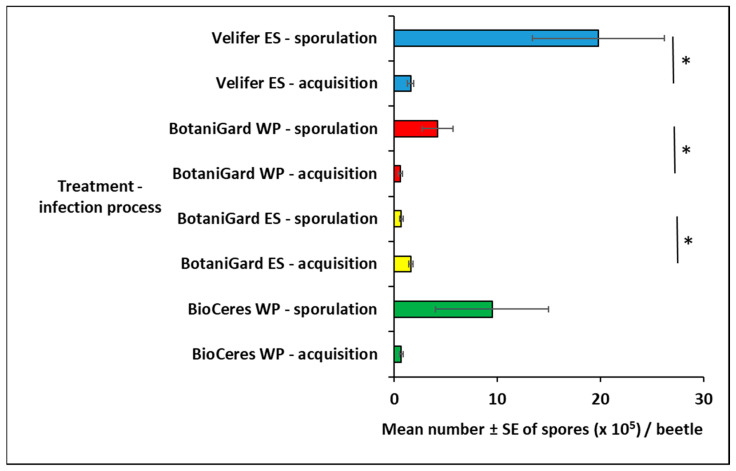
Comparison of the mean number ± SE of spores per TSHB adult acquired during dipping in *Bb*-based fungal products (acquisition) and spore production (sporulation) on beetle cadavers at 21 days of incubation after death. Asterisk per product indicates significant difference (Student’s *t*-test, *p* < 0.05) between acquisition and sporulation.

**Table 1 insects-14-00726-t001:** Four *Beauveria bassiana*-based products and their formulation, fungal strain, manufacturer and fungal phenotype. Control is water only (untreated).

Product	Formulation	Strain	Manufacturer	Phenotype
BioCeres	WP	ANT-03	BioSafe Systems USA	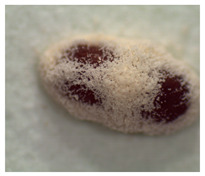
BotaniGard	22WP	GHA	Laverlam International, USA	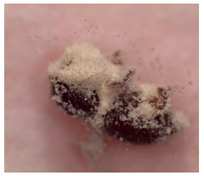
BotaniGard	ES	GHA	Laverlam International, USA	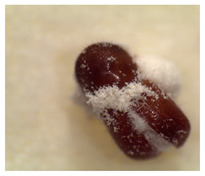
Velifer	ES	PPRI 5339	BASF Corporation, USA	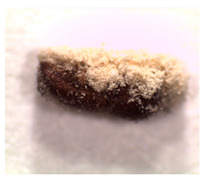
Untreated (control)	N/A	N/A	N/A	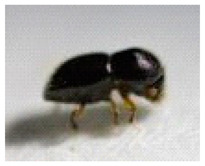

Formulation: WP = Wettable Powder, ES = Emulsion Suspension, N/A = Not Applicable.

## Data Availability

The data are available from the corresponding author if requested.

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
