# Peer review of "Evaluation of Spore Acquisition, Spore Production, and Host Survival Time for Tea Shot-Hole Borer, Euwallacea perbrevis, Adults after Exposure to Four Commercial Products Containing Beauveria bassiana"

_insects, 2023, doi:10.3390/insects14090726_

Round 1

Reviewer 1 Report

This study assessed the potential of four commercial products containing the entomopathogenic fungus Beauveria bassiana for managing and mitigating the spread of the TSHB. Th e authors have tried to support their hypothesis with sufficient arguments and data but the experimental design/scheme only states about some laboratory bioassyas and related data. which makes this study too simple. I would suggest the authors to add some more data on filed efficay to prove the effectiveness of new products compared with the standard one mentioned in this study. Furthermore, somesections of the materials and methods section should be written in detail for reader's convinience. 

Satisfactory

Author Response

Authors response:  We have made it very clear that this manuscript is about the initial screening and assessment of other Beauveria (Bb)-based products, including the standard (BotaniGard ES) against the TSHB under laboratory conditions in the title and throughout the text; and not a field study as the reviewer suggested which is the next step. We also stated this fact in the conclusion of this study. The objective of these studies was to determine if any of the new Bb-based products were more effective compared to the standard (BotaniGard ES) presently being used by the growers for management of ambrosia beetle pests, which now includes the TSHB. We tried to make the Materials and Methods section as detailed as possible for the reader’s convenience.

Reviewer 2 Report

Euwallacea perbrevis is an invasive ambrosia beetle that was introduced to southern Florida over 10 years ago. This introduced pest beetle live inside excavated galleries of host trees and acts as vectors of several phytopathogenic fungi that cause Fusarium branch dieback on avocado trees. The peculiar habit and behavior of this pest challenge for the use of conventional chemical control. The manuscript submitted by Chavez et al. reports the results from a pioneer study to compare the potential of four commercial B. bassiana products (two WP and two ES formulations) for use in the pest control. The study was carried out in a technically sound manner, generating robust data on the number of conidia acquired per adult beetle for each formulation, the survival time per adult beetle after exposure to each formulation, and the epizootic potential per adult cadaver based on the secondary spore production of each registered B. bassiana strain on the cadavers. This well-written manuscript is timely, and referable for our colleagues to develop management strategy against the pest beetle based on fungal insect pathogens.

Author Response

Authors response:  We appreciate the thorough and encouraging comments about our pioneer study having an impact in developing a management strategy for using Bb-based products against the TSHB beetle as stated by Reviewer 2. Reviewer 2 said: “This well-written manuscript is timely, and referable for our colleagues to develop management strategy against the pest beetle based on fungal insect pathogens.”

Reviewer 3 Report

Chavez et al., present the results of a laboratory study on spore acquisition, spore production and survival time of a beetle pest exposed to four formulations of fungal insecticides. The findings will be of interest to readers of Insects. The manuscript requires improvement before it would be suitable for publication.
1. The title needs rewording as the fungi wasn't exposed to the insect, but vice versa, i.e. the insect was the target. I suggest: Spore acquisition, spore production and virulence of commercial formulations of Beauveria bassiana in the tea shoot hole borer Euwallacea perbrevis.
2. Table 1. Suggest you write the words wettable powder and emulsion suspension rather than the codes WP and ES.
3. L183. Delete "Eppendorff"
4. Section 2.6. How did you ensure that the data met the assumptions of normality and equality of variances required for ANOVA and t-tests?
5. Fig 1. y-axis label should include 10e5 as a superscript (as shown in Fig 3)
6. Fig 2. Are these means? As the text mentions ST50 values (i.e., medians)
7. Section 3.3 (and throughout the manuscript) - the term secondary cycling implies transmission which was not measured. I think "spore production" is a more accurate and clearer term.
8. Section 3.4. Modify title to "Comparison of inoculum spore acquisition by dipping verses cadaver sporulation."
9. L283. .... demonstrate the efficacy....
10. L289. Delete the term secondary (which implies transmission has happened). You measured the primary production of spores on a treated insect, rather than the production of spores in subsequent transmission cycles.
11. L297-98. This implies that insects differed in the ability to acquire spores. This was not the case. The four formulations differed in their ability to attach to and contaminate the beetles.
12. Section 4.2. The authors suggest that survival time was due to differences in the DOSE of spores acquired from each formulation, i.e. the number of spores that contaminated each insect. You may have evidence for a correlation here, but you have not demonstrated causation (as far as I could determine). Differences in survival could be related to the virulence phenotype of the active ingredients.
13. Discussion. The Discussion is excessively long and broad ranging. I think the authors should focus on their results, rather than speculating on a diversity of other aspects that affect the efficacy of fungi as biological insecticides. The text could be reduced by 50% without a reduction in relevance. For example, L404-419 is text concerning what might happen in field experiments (not covered by the study).
14. Similarly, the conclusions section is not primarily focused on the findings of the study but speculates on future issues.

Author Response

  1. The title needs rewording as the fungi wasn't exposed to the insect, but vice versa, i.e. the insect was the target. I suggest: Spore acquisition, spore production and virulence of commercial formulations of Beauveria bassiana in the tea shoot hole borer Euwallacea perbrevis

Authors response:  We disagree, because spore acquisition and attachment precede the invasion of the epicuticle of the beetle; therefore, only after the invasion and infection process will the spores be “in” the hemolymph.  After the hemolymph is invaded with blastospores, colonization will take place ultimately culminating in the secondary production of conidia on the host cadaver. We revised the suggested title slightly…

Our revision:

Evaluation of spore acquisition, spore production, and host survival time for tea shot-hole borer, Euwallacea perbrevis, adults after exposure to four commercial products containing Beauveria bassiana.

  1. Table 1. Suggest you write the words wettable powder and emulsion suspension rather than the codes WP and ES.

Authors response: The codes WP, ES and N/A are footnoted on Table 1.

  1. L183. Delete "Eppendorff"

Authors response:  We agree, and “Eppendorf” was removed; “safe-lock” was added.

  1. Section 2.6. How did you ensure that the data met the assumptions of normality and equality of variances required for ANOVA and t-tests?

The individual beetles were randomized when placed in moistened filter chambers for transport. Upon arrival, beetles in each chamber, were then chose at random and placed separately in different bioassay avocado bark plug bioassays. Each set of bioassays were placed randomly with all treatments in a styrofoam holder on a tray, so that all treatments were assessed as a block in the growth chamber. Each block per tray was rotated post assessment day, so that were no biases per treatment. All treatments contained on a tray / block were rotated within the growth chamber shelves. All trays were maintained in total darkness, except when being assessed. Data were also transformed to remove any zeros from the data set and minimize any biases for either ANOVA or t-tests. Percentages were square root + 0.01 arcsine transformed prior to being analyzed using an ANOVA. If the ANOVA indicated significance amongst treatments, then the means per treatment were separated using the LSD test.

  1. Fig 1. y-axis label should include 10e5 as a superscript (as shown in Fig 3)

Authors response:  We agree; fixed as shown in Figure 3

  1. Fig 2. Are these means? As the text mentions ST50 values (i.e., medians)

Authors response: Yes, the graph in Figure 2 is “means”, because the survival time of the beetles per treatment was statistically analyzed using an ANOVA, with mean values being compared using an LSD test (α = 0.05). The statistical results from the Kaplan-Meier survival analysis for the median survival time for 50 % of the population tested (ST50) are only written in the text (no graph shown).

  1. Section 3.3 (and throughout the manuscript) - the term secondary cycling implies transmission which was not measured. I think "spore production" is a more accurate and clearer term.

Authors response:  We agree; the subsection 3.3. as well as the text have been rewritten to reflect “secondary spore production” as suggested.

  1. Section 3.4. Modify title to "Comparison of inoculum spore acquisition by dipping verses cadaver sporulation."

Authors response:  Modified as suggested.

  1. L283. .... demonstrate the efficacy....

Authors response:  Revised.

  1. L289. Delete the term secondary (which implies transmission has happened). You measured the primary production of spores on a treated insect, rather than the production of spores in subsequent transmission cycles.

Authors response:  Deleted “secondary” spore production throughout text as suggested.

  1. L297-98. This implies that insects differed in the ability to acquire spores. This was not the case. The four formulations differed in their ability to attach to and contaminate the beetles.

Authors response:  We agree, and this is written in the text of the conclusion. The differences were only in the formulations because only the TSHB beetles were used in the study. We indicate … However, in contrast to their studies, only the TSHB was assessed in our study; therefore, ….

  1. Section 4.2. The authors suggest that survival time was due to differences in the DOSE of spores acquired from each formulation, i.e., the number of spores that contaminated each insect. You may have evidence for a correlation here, but you have not demonstrated causation (as far as I could determine). Differences in survival could be related to the virulence phenotype of the active ingredients.

Authors response:  We disagree, there is no indication in this section that the survival of the TSHB beetles was related to the DOSE, because all beetles were dipped in the same concentration per well; only the adherence of the spores per formulation of the Bb-based product is suggested for the differences in survival as well as the virulence per Bb strain per product. Other physiochemical aspects per TSHB that may inhibit or enhance the germination of spores are noted as well.

  1. Discussion. The Discussion is excessively long and broad ranging. I think the authors should focus on their results, rather than speculating on a diversity of other aspects that affect the efficacy of fungi as biological insecticides. The text could be reduced by 50% without a reduction in relevance. For example, L404-419 is text concerning what might happen in field experiments (not covered by the study).

Authors response:  We disagree, the information in the Discussion is necessary and pertinent for understanding the next step in assessing these products in the field.

  1. Similarly, the conclusions section is not primarily focused on the findings of the study but speculates on future issues.

Authors response:  We disagree, the information in the Conclusion is necessary and pertinent for understanding the next step in assessing these products in the field. It was also suggested by one of the co-authors to indicate future studies using these formulations because this is an ongoing project.

Reviewer 4 Report

This study assessed the potential of four commercial products containing the entomopathogenic fungus Beauveria bassiana. The manuscript is well prepared. I require one major revision and two minor.

Major point:

 1 Regarding a statistical test for the percentages of mycosed beetles, I suggest employing logistic regression.

Minor points: 

1. In the Discussion part, the authors describe the possibility of killing progeny by horizontal transmission from the infected female dying in the gallery after boring. However, in conclusion, there is a description that “The main limitation of EPFs is they cannot control ambrosia beetles breeding inside the trees. (L436-437)” Sounds inconsistent.

2. In Materials and Methods, the words “random” and “randomly” are used. It needs to mention how to randomize. Or replace these with "arbitrary" or "haphazardly" if randomization was not employed.

Author Response

Major point:

 1 Regarding a statistical test for the percentages of mycosed beetles, I suggest employing logistic regression.

Authors response:  We appreciate the suggestion; but feel the LSD t-test is sufficient for the data collected from separate groups/treatment.

Minor points: 

1.In the Discussion part, the authors describe the possibility of killing progeny by horizontal transmission from the infected female dying in the gallery after boring. However, in conclusion, there is a description that “The main limitation of EPFs is they cannot control ambrosia beetles breeding inside the trees. (L436-437)” Sounds inconsistent.

Authors response:  We disagree, it is not inconsistent. There is limitation on the beetles already breeding inside the tree that were never exposed to the EPF (1st generation of beetles prior to EPF being applied to the tree). However, the beetles that are flying and landing on the tree exposed to the EPF sprayed on the trunk of the tree prior to excavating a gallery in the tree (second generation); this generation of beetles are susceptible to the EPF prior to and post excavation of the gallery. The horizontal transmission from the infected female dying and infecting any progeny with EPF has been observed inside the galleries.

  1. In Materials and Methods, the words “random” and “randomly” are used. It needs to mention how to randomize. Or replace these with "arbitrary" or "haphazardly" if randomization was not employed.

Authors response:  We disagree, we believe it is clearly written how the sample beetles were randomly selected. The word “actively moving” has been inserted in section 2.1.

Round 2

Reviewer 1 Report

The manuscript looks in a good shape but still needs correction of grammar and typos mistakes here and there in text. it can be accepted with a minor revision.

Reviewer 3 Report

The manuscript has been improved.

Minor editing.